# Spectrin binding motifs regulate Scribble cortical dynamics and polarity function

Batiste Boëda[1,2], Sandrine Etienne-Manneville[1]*

[1]Cell Polarity, Migration and Cancer Unit, Institut Pasteur - CNRS UMR 3691, Paris, France; [2]INSERM, Paris, France

**Abstract** The tumor suppressor protein Scribble (SCRIB) plays an evolutionary conserved role in cell polarity. Despite being central for its function, the molecular basis of SCRIB recruitment and stabilization at the cell cortex is poorly understood. Here we show that SCRIB binds directly to the CH1 domain of β spectrins, a molecular scaffold that contributes to the cortical actin cytoskeleton and connects it to the plasma membrane. We have identified a short evolutionary conserved peptide motif named SADH motif (SCRIB ABLIMs DMTN Homology) which is necessary and sufficient to mediate protein interaction with β spectrins. The SADH domains contribute to SCRIB dynamics at the cell cortex and SCRIB polarity function. Furthermore, mutations in SCRIB SADH domains associated with spina bifida and cancer impact the stability of SCRIB at the plasma membrane, suggesting that SADH domain alterations may participate in human pathology.

## Introduction

The protein SCRIB has been implicated in a staggering array of cellular processes including polarity, migration, proliferation, differentiation, apoptosis, stemcell maintenance, and vesicle trafficking (*Humbert et al., 2008*). SCRIB is a membrane associated protein localizing at cell junctions (*Navarro et al., 2005*; *Qin et al., 2005*). Alteration of SCRIB localization at the membrane often mimics SCRIB loss-of-function phenotype (*Zhan et al., 2008*; *Cordenonsi et al., 2011*; *Elsum and Humbert, 2013*) and SCRIB aberrant accumulation in the cytosol strongly correlates with poor survival in human cancers (*Nakagawa et al., 2004*; *Navarro et al., 2005*; *Gardiol et al., 2006*; *Kamei et al., 2007*; *Ouyang et al., 2010*; *Pearson et al., 2011*; *Feigin et al., 2014*). Despite the crucial importance of SCRIB subcellular localization, the molecular basis of SCRIB recruitment and stabilization to the cell cortex is not fully understood.

SCRIB has a tripartite domain organization that consists of an N-terminal region composed of leucine-rich repeats (LRR), PDZ domains and a C-terminal region with no identified protein domain. While the LRR region is crucial for membrane targeting (*Legouis et al., 2003*; *Zarbalis et al., 2004*; *Zeitler et al., 2004*), mutations affecting the PDZ and C-terminal regions have also been reported to affect SCRIB subcellular localization (*Robinson et al., 2011*; *Lei et al., 2013*). The LRR and PDZ domains are highly conserved between *Drosophila* and human (60% of homology). The C-terminal region displays a poor conservation between the two species (13%) (*Figure 1—figure supplement 1*) and its function remains unclear. This is surprising considering that about 40% (5/12) of all the pathological germline mutations identified in the mouse and human *Scrib/SCRIB* genes lies within the C-terminal domain of the protein (*Murdoch et al., 2003*; *Zarbalis et al., 2004*; *Wansleeben et al., 2010*; *Stottmann et al., 2011*; *Lei et al., 2013*).

Here we show that the C-terminal part of SCRIB contains three spectrin binding motifs which are crucial for SCRIB cortical dynamics and polarity function.

*For correspondence: setienne@pasteur.fr

**Competing interests:** The authors declare that no competing interests exist.

**eLife digest** Proteins found in cells often have more than one role. Scribble is one such multi-tasking protein that is found in a diverse range of species, including fruit flies and humans. Although Scribble commonly helps to ensure that the components of a cell are in their correct locations, its exact roles vary between species. To perform its role well, Scribble itself must localize to the cell cortex—the inside surface of the cell membrane—at the regions where cells connect to one another. How this localization occurs is not fully understood; and defects in the human form of Scribble have been linked to diseases including spina bifida and cancer.

Much of the Scribble protein is very similar across different species, but the fruit fly and human version of the protein have large differences in their 'C-terminal region' that makes up one end of each protein. Boëda and Etienne-Manneville now show that in humans and other animals with backbones—but not in fruit flies—the C-terminal region of Scribble contains three repeats of a sequence called the SADH motif. These motifs can bind to proteins called beta spectrins, which connect the cell's outer membrane to the scaffolding-like structure inside the cell that provides support.

Mutations that alter the SADH motif interfere with Scribble's ability to bind to the scaffolding, and alters Scribble localization at cell–cell contacts or the cell cortex. Boëda and Etienne-Manneville also found that some mutations linked to spina bifida and cancer affect the SADH motif, suggesting that this motif has a wider role in disease.

While the abnormal localization of Scribble inside cells is frequently observed in particularly difficult to survive cancers, the molecular mechanism that causes Scribble to fail to localize to the cell periphery is still poorly understood. Boëda and Etienne-Manneville's work establishes the beta spectrin family of proteins as regulators that stabilize Scribble at the cell cortex and suggests that Scribble-associated diseases might depend on the integrity of the spectrin network.

## Results and discussion

### SCRIB C-terminal domain interacts directly with spectrins

We have previously reported that over-expression of the C-terminal part of SCRIB impedes the polarized orientation of the centrosome during astrocyte migration (*Osmani et al., 2006*). The same SCRIB fragment consisting of the last 407 amino acids was used as a bait to screen exhaustively a human fetal brain cDNA library in a two-hybrid screen. Nine independent clones encoding β2-spectrin (*SPTBN1* gene) and seven independent clones encoding β3-spectrin (*SPTBN2* gene) were recovered (*Figure 1—figure supplement 2*). The overlapping sequences of these clones map to the calponin homology one (CH1) domain of spectrins. GST-SCRIB proteins including the C-terminal fragment (1223–1630aa) or the C1 fragment (1223–1424aa) bound endogenous β2 spectrin from cell extracts (*Figure 1A,B*). In contrast, the C2 fragment (1425–1630) did not interact with spectrin (*Figure 1B*). These interactions were also confirmed by co-immunoprecipitation in HEK cells (*Figure 1C*).

Insertion of the alternative exon 36 brings an additional 25aa sequence within the SCRIB C2 protein fragment and forms a long SCRIB isoform (*Figure 1A*). In contrast to the short GFP-C2 fragment (C2 − exon36) domain, the long GFP-C2 isoform (C2 + exon36) strongly interacted with spectrin Flag-CH1 (27–167aa), suggesting that the exon 36 encodes a putative β-spectrin binding sequence (*Figure 1C*). Sequence analysis of the C1 fragment revealed the presence of two 25aa sequences displaying strong similarity with exon 36 sequence. Single deletion of each repeated motif impaired GST-C1 binding to the GFP-tagged CH1 domain of spectrin. The deletion of both repeats totally abolished spectrin binding (*Figure 1D*). GST fusion protein including the 25aa peptide corresponding to the exon 36 as well as the two other similar repeat motifs identified in SCRIB C1, but not GST alone, interacted robustly with GFP tagged CH1 spectrin domain (*Figure 1E*).

We then performed co-localization experiments. In bronchial epithelial cell line (16HBE) SCRIB and both β-spectrins colocalized to adherens junctions (*Figure 1F*). In contrast, in HeLa cells which do not express endogenous cadherin (*Lock and Stow, 2005*), GFP-SCRIB was mainly cytosolic. In these cells, exogenously expressed RFP-CH1 domain of β2 spectrin localises into filamentous structures and

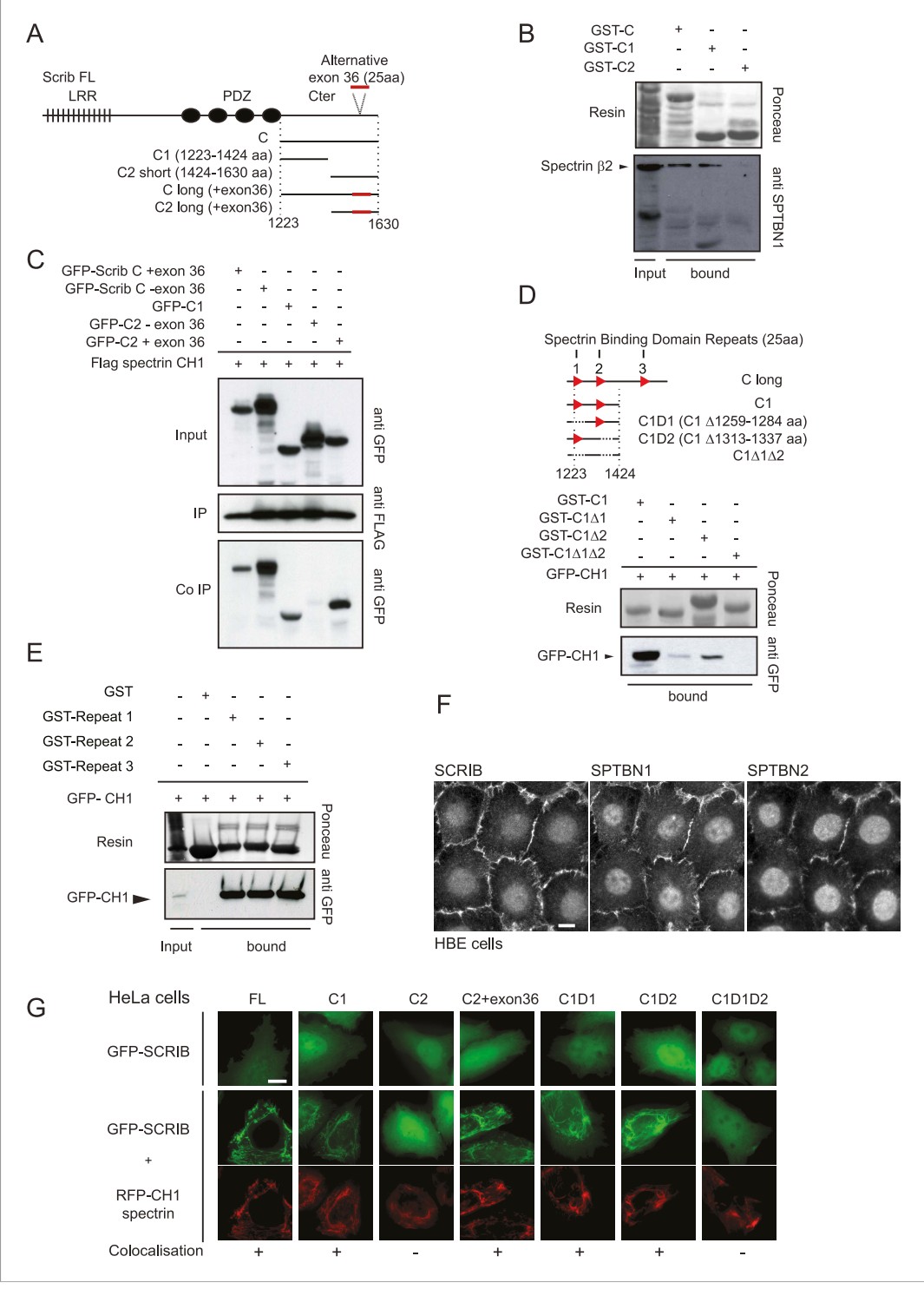

**Figure 1**. Dissection of the SCRIB β-spectrin interaction. (**A**) Schematic representation of SCRIB and different C-terminal constructs used in this study. (**B**) GST-C, GST-C1 and GST-C2 pull-down on astrocytes cell extract. Samples were analyzed by ponceau staining and immunoblotting using the indicated antibody. (**C**) Immunoprecipitation was performed with anti Flag antibody using HEK293 cell lysates co-expressing Flag CH1 spectrin domain with indicated GFP-SCRIB constructs. Samples were analyzed by immunoblotting using the indicated antibodies. (**D**) Schematic representation of the C1 SCRIB constructs with internal deletions of putative spectrin binding motif sequences. GST-C1, GST-Δ1, GST-Δ2, GST-Δ1Δ2 pull-down assay on HEK293 cell lysates expressing GFP-CH1 spectrin domain. Samples were analyzed by immunoblotting using anti GFP. (**E**) GST and GST-SCRIB repeat 1, 2 and 3 pull down

*Figure 1. continued on next page*

*Figure 1. Continued*

assay on HEK293 cell lysates expressing GFP-CH1 spectrin domain. Samples were analyzed by ponceau staining and immunoblotting using the indicated antibody. (**F**) Immunofluorescence images of 16HBE monolayers fixed and stained for SCRIB, SPTBN1 (spectrin β2) and SPTBN2 (spectrin β3). (**G**) HeLa cells were singly transfected with indicated GFP-SCRIB constructs (upper panel) or in combination with spectrin RFP-CH1 domain (bottom panels). Images are representative of at least three independent experiments. Bars, 10 μm.
The following figure supplements are available for figure 1:

**Figure supplement 1**. SCRIB C-terminal sequence alignment.

**Figure supplement 2**. Schematic representation of SCRIB, β2-spectrin and β3 spectrin proteins.

induces the relocalisation of GFP-SCRIB to these structures (*Figure 1G*). In these conditions the GFP-SCRIB lacking putative spectrin binding domains remained cytosolic (*Figure 1G*). Together our results indicate that SCRIB interacts directly with β-spectrins via three spectrin binding motifs with one of them included in an alternative exon.

## SCRIB spectrin binding motifs bind to the CH1 domains of the β spectrin familly

The β2 spectrin N-terminal actin binding region is composed of a tandem calponin homology domain designated CH1-CH2 (*Figure 2A*). GST coupled C1 SCRIB fragment interacted with GFP tagged β spectrin CH1 but neither with CH2 nor with CH1-CH2 tandem domains expressed in HEK cells (*Figure 2B*). In agreement with this observation RFP-CH1 but not RFP-CH1-CH2 β2 spectrin constructs strongly colocalises with F-actin and GFP-C1 SCRIB fragment in HeLa cells. Furthermore, C1 SCRIB fragment interacted with actin-associated CH1 domain (*Figure 2D*) suggesting that SCRIB as a preferential affinity with actin-associated β2 spectrin. Structural studies have shown that the CH1–CH2 tandem domains can switch between an open conformation where the CH1 domain binds to F-actin robustly (*Way et al., 1992*) and a closed conformation in which the CH1 and CH2 domains are closely apposed and display a weak F-actin affinity (*Sjöblom et al., 2008*; *Galkin et al., 2010*).

Calponin homology domains are widespread in the human genome (157 domains referenced in SMART database) and are classified in three classes CH1, CH2 and CH3. We could not detect any interaction with the β2 spectrin CH2 domain (*Figure 2B,F*) or the CH3 domain of Vav3 or IQGAP proteins (*Figure 2E,F*). However, we found that at physiological salt stringency a SCRIB GST-SADH3 bound to the CH1 domains of spectrins β, β2, β3, β5 and α-actinin 2 (*Figure 2E,F*). At high salt stringency (500 mM NaCl), only the spectrin β, β2 and β3 displayed binding with the GST-SADH3 motif (*Figure 2F*). These results indicate that SCRIB has a preferential affinity with the CH1 domain from the close homologues spectrin β, β2 and β3.

## Identification of the SADH (SCRIB ABLIMs DMTN Homology) motif

Phylogenetic analysis revealed that the spectrin binding repeats are conserved in all vertebrate SCRIB sequences tested (primate, rodent, bird, amphibian and fish) but are not present in *Drosophila* or *Caenorhabditis elegans scribble/LET-413* (*Figure 3—figure supplement 1A*). Alignment of all spectrin binding repeats revealed a conserved [+]-X-X-Y-[+]-X-φ-A-A-φ-P sequence (*Figure 3A*). We examined the consequences of alanine or glycine substitutions in SCRIB repeat 3. Mutations of both positively charged residues did not noticeably affect the binding to spectrin CH1 domain. However, mutations of the tyrosine, the alanine doublet or the proline of the repeat 3 severely weakened or completely abolished GST-repeat 3 binding to spectrin CH1 domain (*Figure 3B*).

A search in the human protein database (Swiss-prot) for the simplified Y-[KR]-X-[FL]-A-A-[ILV]-P motif sequence identified four other proteins displaying a perfect fit with the SCRIB motif consensus: DMTN (dematin/Band 4.9) and the three member of the ABLIM family of protein (Actin Binding LIM domain containing proteins) (*Figure 3C*). We found that the DMTN and the ABLIMs motifs retained GFP tagged spectrin CH1 domain from cell extract (*Figure 3D*). Interestingly DMTN is a constituent of the spectrin-actin junctional complex and was known to bind directly to spectrin (*Koshino et al., 2012*). Phylogenetic analyses of the three ABLIMs showed that the spectrin CH1 binding motif was

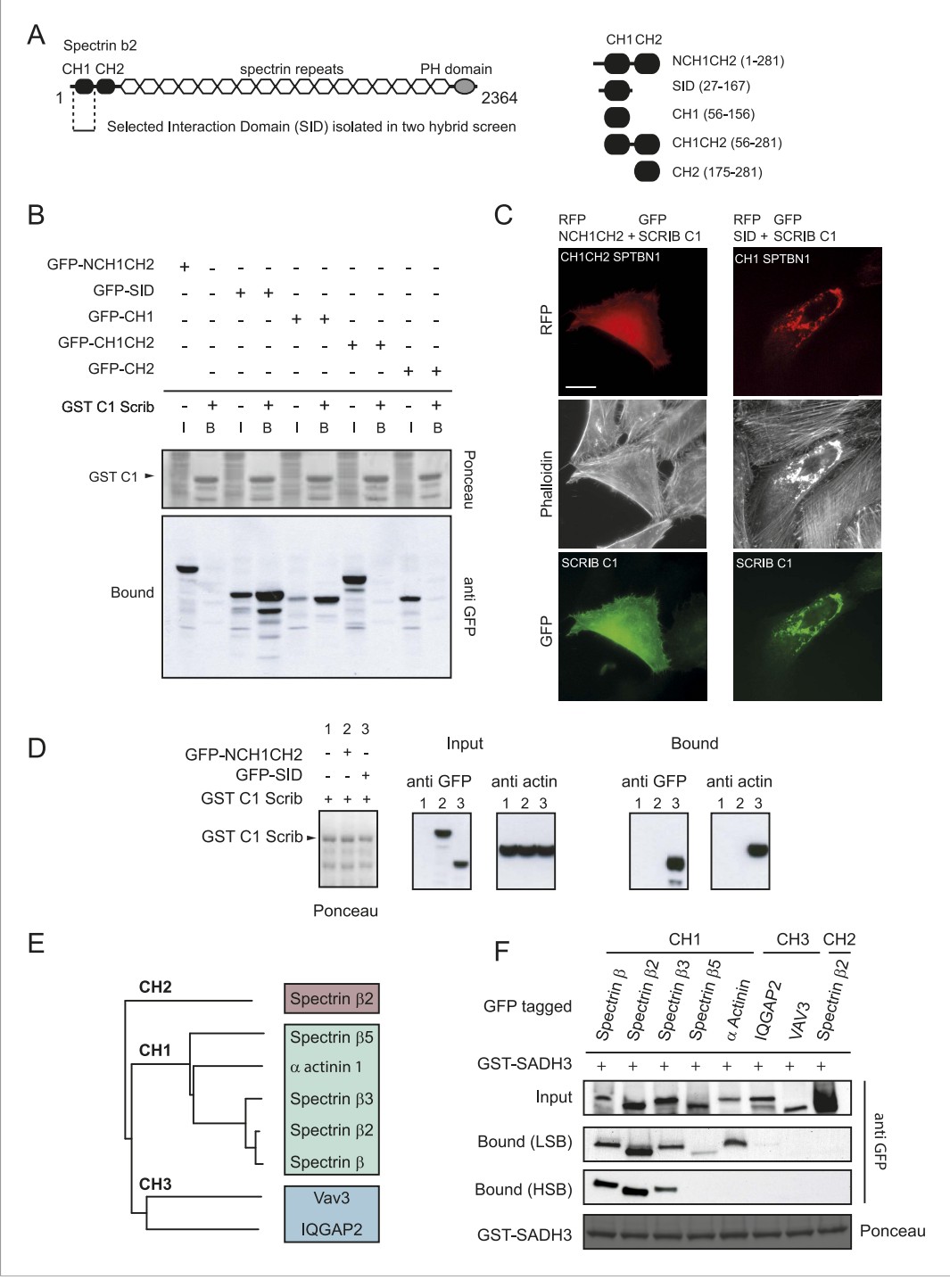

**Figure 2**. SCRIB spectrin binding motifs bind to CH1 domains of the β spectrin family. (**A**) Schematic representation of full length human β2 spectrin protein (left) and different β2 spectrin deletion constructs used in the study (right). (**B**) GST pull down using a GST-SCRIB C1 resin from HEK293 cell lysates expressing indicated GFP-tagged β2 spectrin CH domains. Samples were analyzed by ponceau staining and immunoblotting using anti GFP. (**C**) Fluorescence images showing HeLa cells expressing GFP-SCRIB C1 construct together with RFP-NCH1CH2 or SID β2 spectrin domains and stained with phalloidin. (**D**) GST-SCCRIB C1 pull down from HEK293 cell lysates expressing indicated GFP tagged β2 spectrin CH domains. Samples were analyzed by ponceau staining and immunoblotting using anti GFP. (**E**) Phylogenetic analysis of the Calponin Homology (CH) domains type1, 2 and 3 used in this study. (**F**) GST pull down from HEK293 cell lysates expressing the indicated CH domains using a GST-SCRIB SADH3 resin. Pull down was done in low or high stringency conditions (LSB: Low Salt Buffer, HSB: High Salt Buffer).
*Figure 2. continued on next page*

*Figure 2. Continued*

Ponceau staining indicates the relative amount of GST tagged proteins bound to the resin. Samples were analyzed by ponceau staining and immunoblotting using anti GFP. Bars, 10 μm.

conserved in all vertebrate ABLIMs but could not be detected in invertebrate ABLIMs proteins (*Figure 3—figure supplement 1B*). We also observed that the noncanonical motifs identified in MyoX, MPP and Afadin did not bind to spectrin CH1 domain (*Figure 3D*). Altogether those results indicate that the consensus sequence Y-[KR]-X-[FL]-A-A-[ILV]-P is a spectrin binding motif specific of the vertebrate sub-phylum and present in at least five different genes in the human genome. We

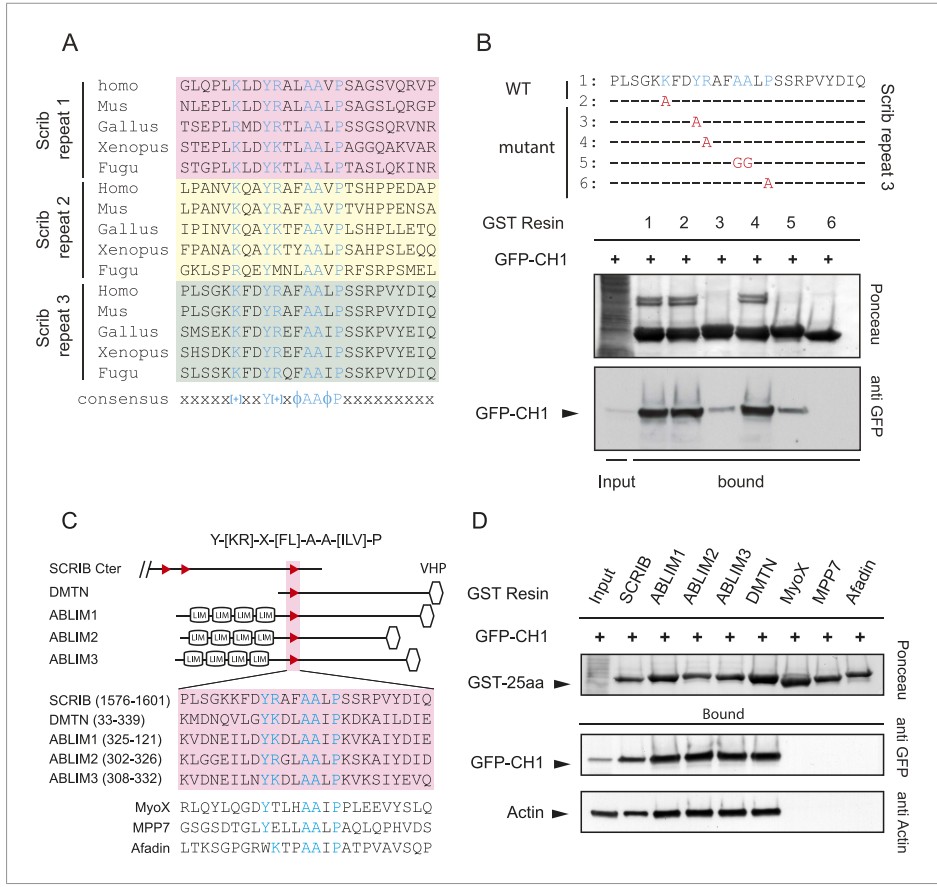

**Figure 3**. The SADH motif (SCRIB, ABLIM, dematin homology motif). (**A**) Alignment of vertebrate SCRIB spectrin binding repeats 1 (pink), 2 (yellow) and 3 (green). Conserved residues are shown in blue. [+] indicates positively charged amino acids and Φ represents hydrophobic residues. (**B**) Alanine or glycine substitutions (in red) performed on SCRIB repeat 3 conserved residues. WT and mutants GST-repeat 3 pull down assay on HEK293 cell lysates expressing spectrin GFP-CH1 domain. Samples were analyzed by ponceau staining and immunoblotting using the indicated antibody. (**C**) Schematic representation and sequence of DMTN and ABLIM1, 2 and 3 proteins displaying strong homology with the SCRIB putative spectrin binding motif Y-[KR]-X-[FL]-A-A-[ILV]-P (boxed in pink). Position of the motif in the protein is indicated in brackets in aa. MyoX, MPP7 and Afadin sequences with noncanonical motifs consensus are shown on the bottom. VHP (Villin Hedapiece domain). (**D**) Pull down assay with GST resin fused to 25aa motif from indicated proteins performed in HEK293 cell lysates expressing spectrin GFP-CH1 domain. Samples were analyzed by ponceau staining and immunoblotting using the indicated antibody.

The following figure supplement is available for figure 3:

**Figure supplement 1**. SADH domains in vertebrates.

named this sequence the SADH motif (SCRIB ABLIMs DMTN Homology). No homology between the SADH motif sequences and the previously characterized 10 kDa spectrin-actin-binding domain of the band 4.1 family of proteins could be detected (*Gimm et al., 2002*).

## SADH motifs are necessary for SCRIB membrane stability

We then investigated the effect of SADH motif mutations (P > A mutant described in *Figure 2B*) on SCRIB cortical localization. We determined the Cortical Localisation Index (CLI, *Figure 4—figure supplement 1*), defined as the ratio between the mean GFP fluorescence intensity in the cortical region and the mean fluorescence intensity in the cytoplasm of the cell. A CLI superior to one indicates that GFP is enriched at the cell cortex while a CLI equal to one shows that the GFP is equally distributed between the cytoplasm and the cortex. GFP alone had a CLI around 1 (0.9) whereas a control GFP-CAAX construct which essentially localized at the cell membrane had a CLI of 1.75. Short and long WT SCRIB (−/+ exon36) constructs showed a strong cortical accumulation (CLI = 1.8 and 2 respectively). In contrast the P305L SCRIB mutant that carries a mutation in the N-terminal LRR region which impedes SCRIB plasma membrane localization (*Legouis et al., 2003*) was mainly cytosolic (CLI = 1). SCRIB SADH1+2 mutant showed a statistically weaker cortical recruitment (CLI = 1.4) than the WT protein (*Figure 4A*) suggesting that SADH mutations may influence SCRIB recruitment and/or stabilization at the membrane. We assessed SCRIB WT and SADH mutant protein dynamics at the membrane by monitoring FRAP (Fluorescence Recovery After Photobleaching) at cell–cell contact in confluent transiently transfected 16HBE cells. A membrane associated GFP-CAAX showed a fast recovery rate ($t_{1/2}$ = 10 s) and the vast majority of the protein was available for exchange (mobile fraction of 97%) (*Figure 4B,C,D*). In contrast GFP-WT SCRIB exchanged less rapidly ($t_{1/2}$ = 26 s) and had a smaller mobile fraction (60%). SCRIB SADH12 mutant displayed a significantly faster exchange ($t_{1/2}$ = 18 s) than the WT SCRIB protein with 82% of the protein pool available for exchange (*Figure 4B,C,D*). These results indicate that the SADH motifs are important for SCRIB stabilization at the cell cortex.

SCRIB is involved in both polarization and orientation of migrating cells in in vitro scratch 'wound-healing' assay (*Qin et al., 2005*; *Osmani et al., 2006*; *Dow et al., 2007*). Expression of GFP-SCRIB C-terminal constructs and to a lesser extent of GFP SADH3 perturbed centrosome reorientation in migrating astrocytes (*Figure 4E,F*) (*Osmani et al., 2006*). Mutations of SADH domains strongly reduced the ability to disrupt centrosome reorientation indicating that the SADH domains are involved in SCRIB polarity fonction. β2 spectrin depletion moderately but significantly perturbed scratch-induced centrosome reorientation (*Figure 4G*). This effect, likely underestimated because of the incomplete knock down or the compensatory role of other spectrins, indicates a role of spectrin in the control of centrosome positioning. Moreover, spectrin β2 depletion impaired SCRIB Cter ability to disrupt centrosome reorientation (*Figure 4G*). Together, these results suggest that SCRIB interaction with spectrin contributes to SCRIB polarity function.

SCRIB is required for the recruitment and the activation of Cdc42 at the cell front edge leading to the centrosome reorientation (*Osmani et al., 2006*; *Dow et al., 2007*). GFP SCRIB Cter overexpression but not GFP alone significantly reduced Cherry-Cdc42 membrane recruitment at the leading edge (*Figure 4—figure supplement 2*) while SCRIB SADH12 mutant Cter construct had significantly weaker effect, suggesting that SCRIB SADH domains are implicated in the SCRIB-mediated recruitment of Cdc42 at the leading edge of migrating cells.

In addition to its role in polarity, SCRIB has been previously implicated in tight junction (TJ) assembly in intestinal epithelium (*Qin et al., 2005*; *Ivanov et al., 2010*). After calcium switch, SCRIB-depleted Caco-2 cell monolayers showed short and disconnected areas of ZO-1 labeling at the cell–cell contacts, indicative of a significant delay in TJ reassembly (*Figure 4—figure supplement 3A, B,C*). This tight junction phenotype could be partially rescued by the expression of a siRNA resistant WT SCRIB but not by SCRIB SADH12 mutant, suggesting that SCRIB interaction with spectrin plays a role in tight junction assembly (*Figure 4—figure supplement 3D*). Altogether these results strongly suggest that SCRIB SADH motifs control SCRIB dynamics at the cell cortex and are important for polarity and tight junction assembly.

## SADH motif mutation in human pathology

61 missense mutations in the *SCRIB* gene coding sequence have been identified so far (COSMIC, [*Forbes et al., 2015*]). Almost 10% of these mutations (6/61) fall within the *SCRIB* SADH motifs

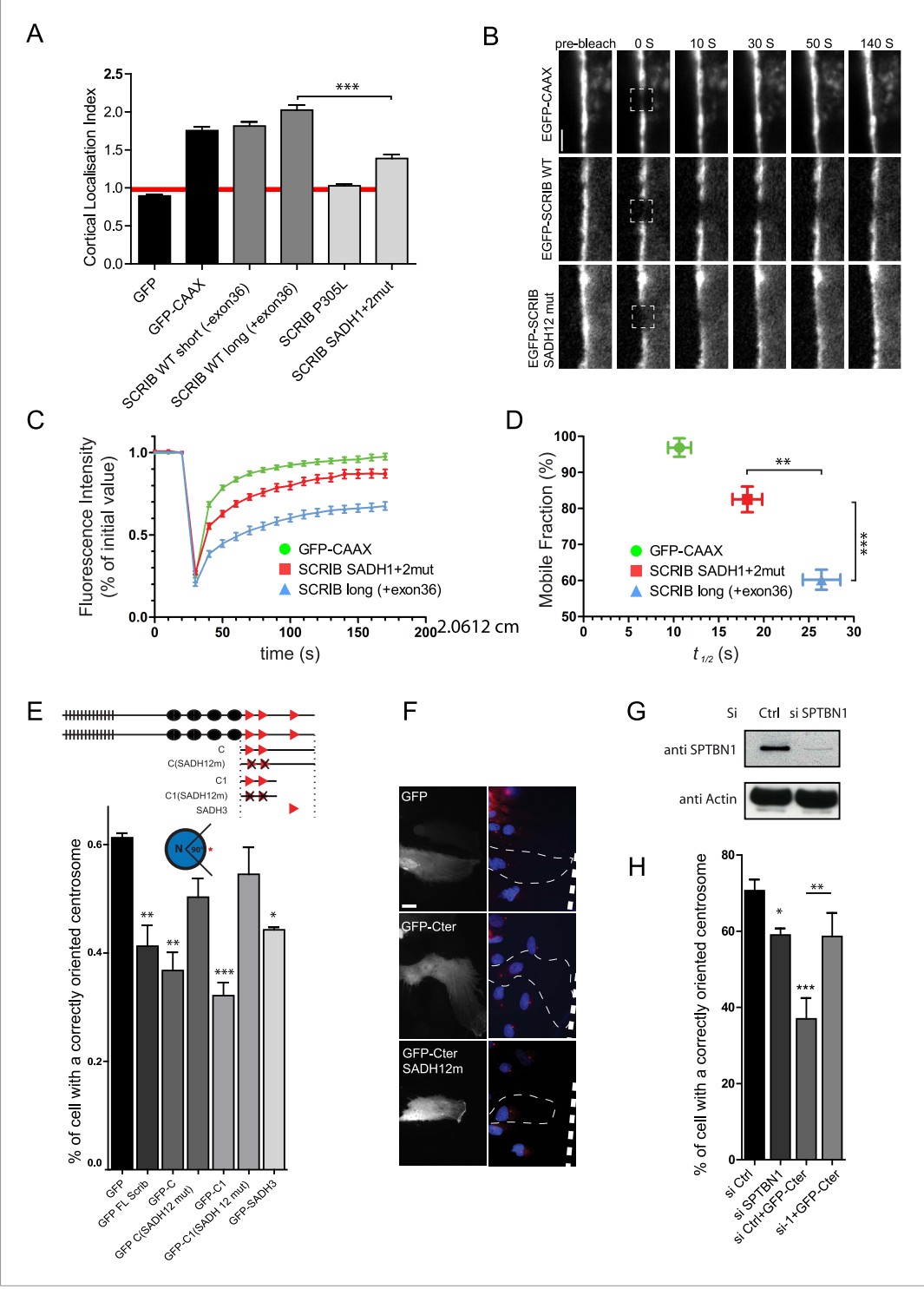

**Figure 4**. SADH motif influence SCRIB membrane stability and is required for SCRIB polarity function. (**A**) 16HBE cells were transiently nucleofected with the indicated GFP constructs and analyzed by live confocal microscopy to calculate their cortical localization index. (n = 50 for each conditions). (**B**) FRAP experiment on 16HBE cells nucleofected with the indicated GFP constructs. High magnification images of adherens junction before and at the indicated time points after photobleaching. (**C**) Quantitative analysis of FRAP from experiments similar to those shown in **B** (n = 30 for each conditions). (**D**) The mobile fraction and $t_{1/2}$ of recovery for each protein were calculated from the recovery curves in **C**. (**E**) Centrosome reorientation in migrating astrocytes expressing the indicated DNA constructs (up). Results shown are means SEM of 3–4 independent experiments, with a total of at least of 150 cells.
*Figure 4. continued on next page*

*Figure 4. Continued*

(**F**) Representative images of astrocytes expressing the indicated constructs and stained with anti-pericentrin (centrosome, red) and Hoechst (nucleus, blue). (**G**) Primary astrocytes were nucleofected with control (Ctrl) or β2 spectrin (si SPTBN1) siRNA and incubated for 72 hr. Protein levels were analyzed by western blots using anti-SPTBN1 antibody and anti-actin. (**H**) Centrosome reorientation assay in migrating astrocytes nucleofected with control or β2 spectrin (SPTBN1) siRNA and microinjected with GFP SCRIB Cter construct. Results shown are means ± SEM of three independent experiments, with a total of at least 100 cells. ***p < 0.001; **p < 0.01; *p < 0.05. Bars, 10 μm.

The following figure supplements are available for figure 4:

**Figure supplement 1**. Cortical localization of SCRIB SADH mutant.

**Figure supplement 2**. SADH domains are implicated in SCRIB-mediated control of Cdc42 localization.

**Figure supplement 3**. SCRIB SADH domain are required for tight junction assembly.

---

(*Figure 5A*), which account for less than 5% of *SCRIB* sequence (75/1657aa). One of the mutations (R1322W) identified in a lung cancer patient directly impacts the core SADH2 consensus motif. In contrast to GST-SADH2 WT sequence, GST-SADH2 R1322W mutant did not bind to the spectrin GFP-CH1 domain pointing to the R1322W mutation as a spectrin binding loss of function mutation (*Figure 5B*). *SCRIB* mutations have also been described in congenital diseases (*Robinson et al., 2011*; *Lei et al., 2013*). In particular, six SCRIB mutations have been recently described in *spina bifida* (*Lei et al., 2013*) and two of those six fall within the SADH2 motif sequence (*Figure 5C*). The A1315T mutation did not affect the ability of a GST-SADH2 to bind to GFP spectrin. In contrast the P1332L mutation noticeably increased GST-SADH2 affinity for GFP spectrin in vitro (*Figure 5C,D,E*). These mutations did not impact significantly SCRIB overall recruitment to the cellular cortex (*Figure 5—figure supplement 1*). However, FRAP experiments showed that GFP R1322W and GFP P1332L SCRIB mutants exchanged more rapidly than the GFP WT SCRIB at the plasma membrane (*Figure 5F*). Surprisingly, P1332L mutation also increased SCRIB exchange at the plasma membrane, suggesting that, in the context of the SCRIB full length molecule, the P1332L mutation prevents rather than increases spectrin binding. We cannot however exclude the possibility that this mutation also affects other yet-unidentified function of the SADH domain. Altogether these observations indicate that mutations of the SADH motifs may participate in human pathology by impacting the stability of SCRIB at the cell cortex.

Dow et al have shown that a transgene encoding the human gene *SCRIB* was able to partially rescue the *Drosophila scribble* mutant (23% survival rescue to adult stage), arguing for an indisputable conserved role of *scribble/SCRIB* during evolution (*Dow et al., 2003*). Nevertheless it is clear that some phenotypic differences exist between *Drosophila* and mouse *Scrib* mutant models like the severity of apico-basal epithelium polarity phenotype, and that the *scribble/SCRIB* genes have undergone some intra-phylum specific adaptation (*Bilder et al., 2000*; *Murdoch et al., 2003*). No phylum specific SCRIB interacting partner has been characterized so far that could account for the differences observed between invertebrates and vertebrates SCRIB function. The vertebrate-specific spectrin binding motifs SADH located inside SCRIB divergent C-terminal region appears to be a good candidate to bear such role.

## Materials and methods

### Yeast two-hybrid analysis

Yeast two-hybrid screening was performed by Hybrigenics Services, SAS, Paris, France (http://www.hybrigenics-services.com). The coding sequence for human protein SCRIB (aa 1224–1630) (GenBank accession number gi: 18141296) was PCR-amplified and cloned into pB27 as a C-terminal fusion to LexA (N-LexA-SCRIB-C) and into pB66 as a C-terminal fusion to Gal4 DNA-binding domain (N-Gal4-SCRIB-C). The constructs were checked by sequencing and used as a bait to screen a random-primed Human Fetal Brain cDNA library. A confidence score (PBS, for Predicted Biological Score) was attributed to each interaction as previously described (*Formstecher et al., 2005*).

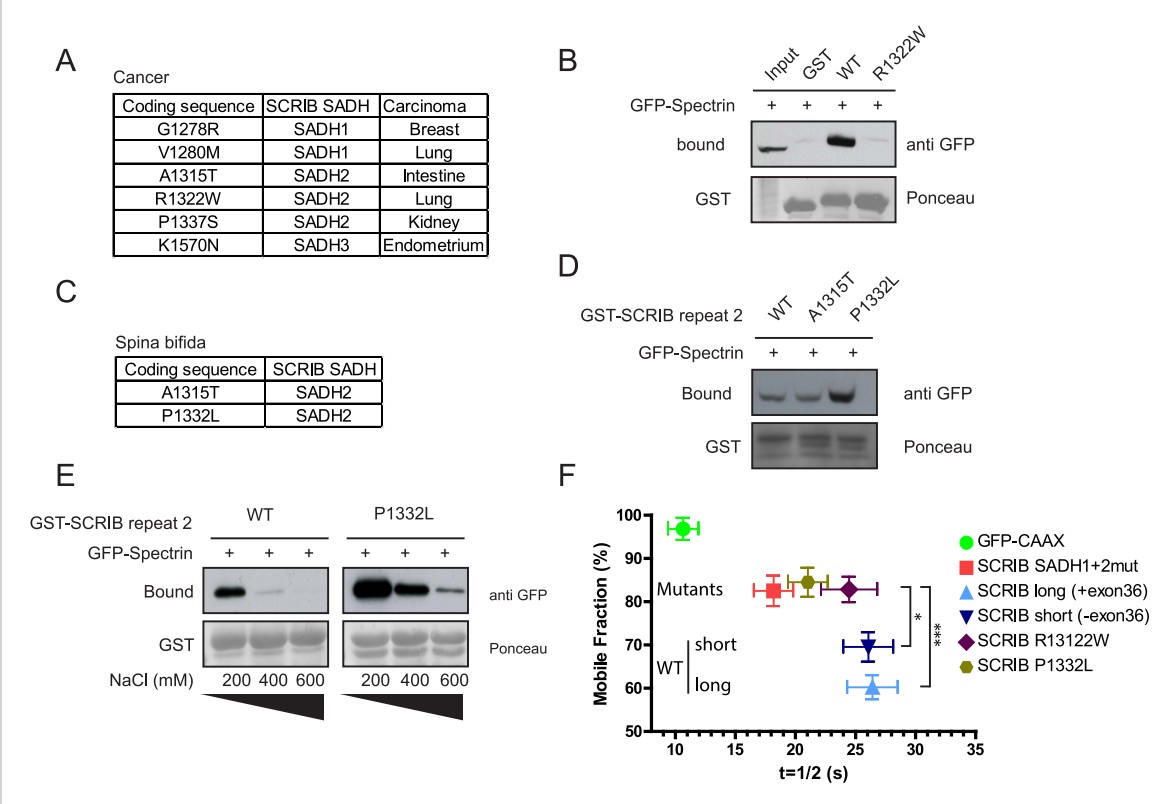

**Figure 5**. SADH motifs mutations in human pathology. (**A**) Table of identified somatic SCRIB SADH motifs mutations in human cancers. Data sourced from COSMIC (http://cancer.sanger.ac.uk/cosmic). (**B**) GST-SCRIB SADH2 WT and SADH2 R1322W pull down assay on HEK293 cell lysates expressing spectrin GFP-CH1 domain. Ponceau staining indicates the relative amount of GST tagged proteins bound to the resin. Samples were analyzed by immunoblotting using anti GFP. (**C**) Table of identified germinal SCRIB SADH motifs mutations in *spina bifida*. (**D**) GST-SCRIB SADH2 WT, A1315T and P1332L pull down assay on HEK293 cell lysates expressing spectrin GFP-CH1 domain. Ponceau staining indicates the relative amount of GST tagged proteins bound to the resin. Samples were analyzed by immunoblotting using anti GFP. (**E**) GST-SCRIB SADH2 WT and P1332L pull down assays similar to **D** performed in indicated salt stringency. (**F**) Quantitative analysis of FRAP experiments in 16HBE adherens junction expressing the indicated GFP constructs (n = 30 for each conditions). The mobile fraction and $t_{1/2}$ of recovery for R1322W and P1332L proteins were calculated from the recovery curves in *Figure 5—figure supplement 1*.

The following figure supplement is available for figure 5:

**Figure supplement 1**. (**A**) 16HBE cells were transiently nucleofected with the indicated GFP constructs and analyzed by live confocal microscopy to calculate their cortical localization index.

## Material

The following reagents were used in this study: Anti SCRIB C-20 (Goat, Santa Cruz Biotechnology, Dallas, TX), Anti-SPTBN1 and E-cadherin (Mouse, BD Transduction Laboratories, San Jose, CA), Anti-SPTBN2 A301-117A (Rabbit, Bethyl, Montgomery, TX), Anti-GFP HRP ab6663 (Abcam, Cambridge, MA), Anti-FLAG HRP clone M2 (Sigma, Saint Louis, MO), Anti-pericentrin PRB-432C (Rabbit, Covance, Princeton, NJ), Anti-Actin AC-40 (Sigma). SiRNA sequences: Non-Targeting control siRNA (luciferase) (UAAGGCUAUGAAGAGAUAC), SPTBN1 (UGAUGGCAAAGAGUACCUCUT), SCRIB-ORF1 (GCACU GAGGAGGAGGACAATT), ORF2 (GAACCUCUCUGAGCUGAUCTT), UTR1 (GUUCUGGCCUGUGA CUAACTT) and UTR2 (GGUUUAAGGAGAAUAAAGUTT) were ordered at Eurofins (France).

## DNA constructs

The SCRIB domains were amplified by PCR from a human SCRIB template provided by Jean Paul Borg and cloned into the NotI-EcoRI sites mammalian expression vectors CB6-N-GFP and pEGFP or the *Escherichia coli* expression vector pMW172-GST. SADH motifs were obtained by annealed oligo

cloning and cloned likewise in the above-mentioned vectors. The SCRIB internal deletions and point mutagenesis were generated using PCR-based site-directed mutagenesis. The spectrin sequence corresponding to the 27–167aa prey clone encompassing the β2-spectrin CH1 domain was cloned in CB6-N-GFP, CB6-N-RFP or CB6-N-Flag.

## Sequence analysis

For sequence homology search the Y-[KR]-X-[FL]-A-A-[ILV]-P motif was blasted on the Pattern Search program at http://www.expasy.org/ (*Sigrist et al., 2010*) against the Homo sapiens Swiss-prot database. Sequence alignments and phylogeny of calponin homology domains were done on the mobile website (*Neron et al., 2009*) and using the seaview program (*Gouy et al., 2010*). Statistical analysis was performed using GraphPad Prism 5.0. SCRIB somatic mutations in cancers were obtained at www.sanger.ac.uk (*Forbes et al., 2015*).

## Protein expression, purification, and resin production

All proteins were expressed in *E. coli* BL21(DE3) Rosetta strain. Bacterial cell pellets were lysed 1 hr at RT in 150 mM NaCl, 50 mM Tris pH8 and 25% sucrose supplemented with 5000 units of lyzozyme (Sigma). Cleared supernatants were mixed for 90 min with glutathione-Sepharose 4B beads (GE Healthcare). The resulting resins were washed three times with PBS containing 200 mM NaCl and 0.1% Triton (buffer A).

## Interaction assays and immunoprecipitation

HEK 293 cells were transiently transfected using the phosphate calcium method. Cell lysates were prepared by scraping cells in lysis buffer 50 mM Tris pH7.5, triton 2%, NP40 1%, 200 mM NaCl with Complete protease inhibitor tablet (Roche, Indianapolis, IN) and centrifuged for 10 min at 13,000 rpm 4°C to pellet cell debris. Soluble detergent extracts were either incubated with GST-SCRIB resins or Anti FLAG coupled protein G-Sepharose (GE healthcare) for 2 hr at 4°C prior to washing three times with buffer A and processed for western blot analysis. For the calponin homology domain binding experiment (*Figure 2B*) the high stringency binding was done in lysis buffer containing 500 mM NaCl.

## Cell culture, nucleofection and immunofluorescence

16HBE cells were maintained in DMEM/F12 medium (Invitrogen), Caco-2 and HEK cells in DMEM supplemented with 10% FBS (Invitrogen) and penicillin (100 U/ml)-streptomycin (100 μg/ml; Invitrogen) at 37°C in 5% $CO_2$. For DNA and siRNA transfection $5 \times 10^6$ 16HBE or Caco-2 cells were nucleofected with DNA (5 μg) using Lonza kitT (program A-23 and B-024 respectively) and nucleofector device, according to manufacturer's protocol. Primary rat astrocytes were prepared as described previously (*Etienne-Manneville, 2006*). For immunofluorescence, cells were fixed in 4% paraformaldehyde, permeabilized with 0.1% Triton X-100 in PBS, and blocked in PBS 10% Serum for 1 hr before incubation with antibodies.

## Scratch assay and calcium switch

For scratch-induced assays, primary astrocytes were seeded on poly-L-ornithine-coated coverslips and were grown in serum to confluence. The medium was changed 16 hr before scratching. Individual wounds (approximately 300 mm wide) were made with a microinjection needle and front row migrating astrocytes were immediately micro-injected with the indicated GFP tagged constructs. Centrosome reorientation was determined as described previously (*Etienne-Manneville and Hall, 2001*). Briefly, 8 hr after the wound, centrosomes located in front of the nucleus of GFP positive front row cells, within the quadrant facing the wound were scored as correctly oriented. In these assays, a score of 25% (astrocytes) represents the absolute minimum corresponding to random centrosome positioning. Calcium Switch was performed as described previously (*Ivanov et al., 2010*). Briefly, Caco-2 cells were incubated for 1 hr in the low-calcium medium and supplemented with 2 mmol/l EGTA before being returned to normal cell culture media (calcium repletion) for indicated times at 37°C. Fixed cells were imaged on a microscope (DM6000 B; Leica) using an HCX Plan Apochromat 40×/1.25 NA oil confocal scanning or HCX Plan Apochromat 63×/1.40 NA oil confocal scanning objective (Leica). Microscopes were equipped with a camera (DFC350FX; Leica), and images were collected with LAS software (Leica).

## Image acquisition and FRAP

16HBE cells grown on MatTek (P35G-1.5-14-C) Petri dishes were analyzed 72 hr after nucleofection. Live cell imaging was performed on a spinning disk confocal microscope Zeiss Axiovert 200 with UltraView ERS (Perkin–Elmer), at 37°C with a Plan-Apochromat X63/1.4 objective. To calclulate the CLI we used the following equation: $CLI = (i/[i + I])/(a/[a + A])$ where (i) represents the pixel intensity contained in a 0.625 μm thick cortical band encompassing the cell edge, (I) the cytoplasmic pixel intensity, (a) the cortical region area and (A) the cytoplasmic area (*Figure 3—figure supplement 1*). The CLI was calculated for each cell on three different Z plan and then averaged. FRAP experiment was performed using the FRAP module of confocal Volocity software (Perkin–Elmer). A 9 μm² square region of interest to be bleached was defined for the FRAP and maximum laser power at 488 nm for one iteration was used to bleach signals. After bleaching, images were taken within the same focal plane at regular intervals (between 3 and 10 s) to monitor fluorescence recovery. After background subtraction the recovery of the GFP signal was measured using ImageJ and fitted using the Prism software and the equation $Y_{(t)} = (Y_{max} - Y_{min}) (1 - e^{2kt}) - Y_{min}$ (*Weisswange et al., 2009*), where Y(t) is the intensity of fluorescence at time t, $Y_{max}$ and $Y_{min}$ are respectively the maximum and minimum intensities of fluorescence post-bleaching and k is the rate constant of recovery. Mobile fraction was determined as $Mf = (Y_{max} - Y_0)/(1 - Y_0)$. All data are presented as the mean ± s.e.m. One-way ANOVA analysis of the variance was followed by the Tukey's multiple comparison post-hoc test. A p value of <0.05 was considered as statistically significant.

## Acknowledgements

We are grateful to JP Borg, Norbert Frey and A El Amraoui for plasmids and thank Jean-Yves Tinevez from the Plate-Forme d'Imagerie Dynamique/Imagopole of Institut Pasteur for technical support. This work was supported by the Institut National du Cancer, l'Association pour la Recherche contre le Cancer, and La Ligue contre le Cancer. B Boëda is supported by Institut National de la Santé et de la Recherche Médicale.

## Additional information

### Funding

| Funder | Author |
|---|---|
| Fondation ARC pour la Recherche sur le Cancer | Sandrine Etienne-Manneville |
| Institut National du Cancer | Sandrine Etienne-Manneville |
| Ligue Contre le Cancer | Sandrine Etienne-Manneville |

The funders had no role in study design, data collection and interpretation, or the decision to submit the work for publication.

### Author contributions

BB, Conception and design, Acquisition of data, Analysis and interpretation of data, Drafting or revising the article; SE-M, Conception and design, Analysis and interpretation of data, Drafting or revising the article

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
