## [Decision Letter]

Thank you for sending your work entitled “Spectrin binding motifs regulate Scribble cortical dynamics and polarity function” for consideration at *eLife*. Your article has been favorably evaluated by Detlef Weigel (Senior editor), Mohan Balasubramanian (Reviewing editor) and two reviewers. You will be pleased to know that the referees find your work of interest and publishable in *eLife*, if appropriate major revisions are carried out.

The Reviewing editor and the reviewers discussed their comments before we reached this decision, and the Reviewing editor has assembled the following comments to help you prepare a revised submission.

The major and minor comments are listed below and all questions seem reasonable, and the experiments suggested by referee 1 in major comments and referee 2 in major (points 1 and 2) and minor comments can be readily done.

Another point raised is the biological relevance (beyond studies in cell lines) of the interaction between Scribble (SADH) and spectrin. I realize this is challenging since the SADH domain does not exist in the *Drosophila* and nematode proteins, where such experiments can be readily done. Is it possible to carry out a relatively straightforward analysis in zebrafish? If so, I would welcome this addition.

However, I hope you can satisfactorily address all the other points raised by the two referees with additional experiments where required.

Reviewer #1:

This manuscript identified a new motif, SADH, in the C-terminal region of Scribble. A significant portion of mutations in Scribble associated human pathology are within the SADH motifs. So this manuscript is addressing an important issue-the function of previously-uncharacterized C-terminus of Scribble. The authors showed that SADH domain directly interacted with Spectrins and that SADH motifs are necessary for membrane stability and polarity function of Scribble. These are very novel and significant findings.

Major points:

1) The data supporting the functional study of SADH motifs in cell polarity is somewhat confusing. Either overexpression or depletion of Scribble resulted in defects in centrosome orientation in the wound-healing assay (26). “Expression of GFP-SCRIB C-terminal and C1 constructs perturbed centrosome reorientation”. Was the purpose of this experiment to compare with the effect of wild-type Scribble overexpression? This result can be interpreted as: 1) they are sufficient to trigger centrosome reorientation as the WT full-length Scribble; or 2) they function as dominant-negative constructs, because PDZ domains of Scribble is known to be important for its polarity function. The results should be explained in a clearer way and information on the control such as overexpression of WT full-length Scribble should be cited. Besides, the phenotype caused by expression of C-terminus of Scrib in the same assay had already been previously described by the same group (26). This data was not cited in the manuscript.

It's unclear how SADH motifs mediate cell polarity. In the wound-healing assay, Scribble is required for Cdc42 activity and localization for centrosome reorientation (26). Are SADH motifs required for CDC42 localization in this assay? If not, what is the new mechanism?

2) Are Spectrins important for centrosome reorientation in the wound-healing assay? Are Spectrins important for cell polarity function medicated by SADH motifs? Do spectrins have any effect caused by expression of Scrib C-terminus or the mutations in this region? The manuscript shows a novel interaction between spectrins and SADH motifs, but lacks functional studies of Spectrins with SADH.

3) The data that both R1322W and P1332L exchanges more rapidly at the plasma membrane in the FRAP experiments argues against the relevance of the SADH-Spectrins binding with Scrib stability, because P1332L binds to Spectrin with a higher affinity and should be more stabilized on the plasma membrane. Could the binding of SADH motifs with Spectrins contribute to a cellular function of Scrib other than cell polarity? *Spina bifida* does not seem to be caused by cell polarity defects/tissue overgrowth, but rather by defects in neural tube development.

Reviewer #2:

The manuscript describes a new binding partner for vertebrate Scribble proteins, the cytoskeletal protein Spectrin, and identifies a new Spectrin-binding motif in Scribble and four other proteins. The authors also demonstrate a role for this motif stabilizing Scribble in the cortex of cultured cells, and investigate a few human binding motif mutants in similar assays.

One weakness is that the authors do not investigate the Scribble binding domain of Spectrin, beyond showing that it is present in Spectrin's actin-binding CH1 domain, or show whether this binding affects Spectrin function.

A second weakness is that the assays do not address the role of this binding in what is Scribble's main known function, the formation of cell junctions. This is largely a biochemical analysis with only a few in vitro assays added.

Nonetheless, while it would be nice to have a bit more on biological relevance, and on the binding motifs in Spectrin, the data presented looks solid, and the authors for the most part draw logical conclusions.

---

## [Author Response]

As reviewer #1 suggested, we investigated Cdc42 recruitment in migrating astrocytes and showed that SCRIB C-terminal WT domain but not the SADH mutant affects Cdc42 leading edge localization. Moreover, we show that β2 spectrin depletion moderately but significantly perturbed scratch-induced centrosome reorientation. Interestingly, spectrin β2 depletion impaired SCRIB Cter ability to disrupt centrosome reorientation. Together, these new results suggest that SCRIB interaction with spectrin contributes to SCRIB function in cell polarization.

As reviewer #2 suggested, we now provide substantial new results concerning SCRIB binding domain in spectrins. These results are now presented in the new Figure 2. We now show that SCRIB binds to the CH1 of the close homologues spectrin β, β2 and β3 and has a preferential affinity with actin-associated CH1 domain. Moreover, the SADH motifs bind specifically to the actin associated CH1 domain subfamily but not to the closely related non actin binding CH2 and CH3 domains.

We have also investigated the role of SCRIB-spectrin binding in the formation of epithelial junctions. Using the “calcium switch” model, we show that the wild type GFP SCRIB, but not the GFP SCRIB SADH mutant can partially rescue the delay of tight junction 2 formation observed in SCRIB-depleted caco-2 cells. These results points towards a role of SCRIB SADH domains in junction formation.

*Another point raised is the biological relevance (beyond studies in cell lines) of the interaction between Scribble (SADH) and spectrin. I realize this is challenging since the SADH domain does not exist in the* Drosophila *and nematode proteins, where such experiments can be readily done. Is it possible to carry out a relatively straightforward analysis in zebrafish? If so, I would welcome this addition*.

Unfortunately we were not able to carry out the zebrafish experiment that you suggested. This is mainly due to the difficulty to establish a quick collaboration with zebrafish lab working on SCRIB.

*However, I hope you can satisfactorily address all the other points raised by the two referees with additional experiments where required*.

Reviewer #1:

*This manuscript identified a new motif, SADH, in the C-terminal region of Scribble. A significant portion of mutations in Scribble associated human pathology are within the SADH motifs. So this manuscript is addressing an important issue-the function of previously-uncharacterized C-terminus of Scribble. The authors showed that SADH domain directly interacted with Spectrins and that SADH motifs are necessary for membrane stability and polarity function of Scribble. These are very novel and significant findings*.

*Major points*:

*1) The data supporting the functional study of SADH motifs in cell polarity is somewhat confusing. Either overexpression or depletion of Scribble resulted in defects in centrosome orientation in the wound-healing assay (*[26]*). “Expression of GFP-SCRIB C-terminal and C1 constructs perturbed centrosome reorientation”. Was the purpose of this experiment to compare with the effect of wild-type Scribble overexpression? This result can be interpreted as: 1) they are sufficient to trigger centrosome reorientation as the WT full-length Scribble; or 2) they function as dominant-negative constructs, because PDZ domains of Scribble is known to be important for its polarity function. The results should be explained in a clearer way and information on the control such as overexpression of WT full-length Scribble should be cited*.

We had initially (26) expressed various fragment of SCRIB in migrating astrocyte to determine which domain(s) of the protein played a role in the control of centrosome positioning. These experiments revealed that SCRIB C- terminal domain acts as a dominant negative construct in centrosome reorientation. The purpose of the present study was thus to determine what is the exact implication of this region. We have clarified this point in the revised manuscript and added the result obtained with FL SCRIB in centrosome reorientation for comparison (Figure 4).

*Besides, the phenotype caused by expression of C-terminus of Scrib in the same assay had already been previously described by the same group (*[26]*). This data was not cited in the manuscript*.

The [26] paper was already cited in the first sentence of the Results section of the manuscript and twice in the Results section.

*It's unclear how SADH motifs mediate cell polarity. In the wound-healing assay, Scribble is required for Cdc42 activity and localization for centrosome reorientation (*[26]*)*. *Are SADH motifs required for CDC42 localization in this assay? If not, what is the new mechanism?*

We now show that SCRIB C-terminal domain overexpression reduces significantly the percentage of cells with correct leading edge Cdc42 recruitment while the SADH mutant construct was significantly less potent to perturb Cdc42 targeting (Figure 4—figure supplement 2). In agreement with the reviewer’s suggestion, these results indicate that the C-terminal effect on cell polarity passes through control of Cdc42 membrane localization and that the SADH motifs are needed for this function.

2) Are Spectrins important for centrosome reorientation in the wound-healing assay?

To address the reviewer’s question, we have tried to knock down several spectrin among the 7 spectrins (2α and 5β) involved in the cortical cytoskeleton. Most of these proteins have proven to be resistant to siRNA-induced depletion. Nevertheless, we obtained an almost complete knock down of SPTBN1, which is the spectrin displaying the more affinity with SCRIB in the two hybrid assay and found that SPTBN1 depletion significantly perturbs centrosome reorientation in migrating astrocytes (new Figure 4). Moreover, it is interesting to note that mouse knockout models of ubiquitously expressed Spectrin αII and βII die in utero due to incomplete neural tube closure and cardiac anomalies, a pleiotropic association of phenotypes reminiscent of the ones observed in *Scrib* mice mutants (Tang, Katuri et al., 2002; Stankewich, Cianci et al., 2011).

*Are Spectrins important for cell polarity function medicated by SADH motifs? Do spectrins have any effect caused by expression of Scrib C-terminus or the mutations in this region? The manuscript shows a novel interaction between spectrins and SADH motifs, but lacks functional studies of Spectrins with SADH*.

We now show that in SPTBN1-depleted cells the microinjection of SCRIB C- terminal does not induce a strong depolarization defect, suggesting that SCRIB C- terminal domain effect on polarity requires SPTBN1 integrity (new Figure 4).

*3) The data that both R1322W and P1332L exchanges more rapidly at the plasma membrane in the FRAP experiments argues against the relevance of the SADH-Spectrins binding with Scrib stability, because P1332L binds to Spectrin with a higher affinity and should be more stabilized on the plasma membrane. Could the binding of SADH motifs with Spectrins contribute to a cellular function of Scrib other than cell polarity?* Spina bifida *does not seem to be caused by cell polarity defects/tissue overgrowth, but rather by defects in neural tube development*.

We agree with the reviewer that the biochemical effect of the mutations and the FRAP experiment data obtained with the SCRIB mutant P1332L are somehow contradictory. We can only speculate that the P1332L mutation increases spectrin binding ability at the SADH2 25aa peptide level, as observed in our binding assay but has the opposite effect in the context of the full length SCRIB molecule. This hypothesis is difficult to test as we cannot produce full length SCRIB protein in vitro (180 kDa) to perform discriminating binding affinity measurement between WT and mutant SCRIB sequences. As the reviewer suggests, the SADH motifs may also contribute to a SCRIB function different than cell polarity, more connected to neural tube development. However SCRIB is not the only polarity gene found mutated in *spina bifida* (CELSR1, FZD6 and PRICKLE1 and 2; Juriloff and Harris, 2012), suggesting that this pathology is connected to polarity defect. The effects of the P1332L mutation are now further discussed in the revised manuscript.

Reviewer #2:

*The manuscript describes a new binding partner for vertebrate Scribble proteins, the cytoskeletal protein Spectrin, and identifies a new Spectrin-binding motif in Scribble and four other proteins. The authors also demonstrate a role for this motif stabilizing Scribble in the cortex of cultured cells, and investigate a few human binding motif mutants in similar assays*.

*One weakness is that the authors do not investigate the Scribble binding domain of Spectrin, beyond showing that it is present in Spectrin's actin-binding CH1 domain, or show whether this binding affects Spectrin function*.

To address this important issue, we first show that SCRIB C1 fragment pulls down the β2 spectrin CH1 domain but not the CH2 or the CH1-CH2 tandem domain (new Figure 2). Structural studies have shown that the CH1-CH2 tandem domains can switch between an open conformation where the CH1 domain binds to F-actinrobustly and a close conformation in which the CH1 and CH2 domains are closely apposed and display a weak F-actin affinity (Sjoblom, Ylanne et al., 2008; Galkin, Orlova et al., 2010). The GFP tagged β2 spectrin CH1-CH2 domain in HeLa cells display poor F-actin colocalization compared to GFP CH1 domain alone (Figure 2), in agreement with the idea that the β2 spectrin CH1-CH2 tandem domain adopt a close conformation when expressed in cells. Furthermore we show that SCRIB C1 region pulls down the β2 spectrin CH1 domain together with actin, suggesting that SCRIB binds to the CH1 domain in an open conformation and does not compete with actin binding (Figure 2).

Finally, we found that SADH3 motif does not bind to CH2 domain from β2- spectrin or CH3 domains from Vav3 or IQGAP proteins but that at physiological salt stringency it interacts with the CH1 domain of spectrins β, β2, β3, β5 and α-actinin 2 (Figure 2). At high salt stringency (500 mM NaCl), only the spectrin β, β2 and β3 displayed binding with the GST-SADH3 motif (Figure 2), indicating that SADH domains have a preferential affinity with the CH1 domain from the close homologues spectrin β, β2 and β3.

In conclusion, this new Figure 2 shows that the SADH motifs bind specifically to the actin binding CH1 domain subfamily but not to the closely related CH2 and CH3 domains.

*A second weakness is that the assays do not address the role of this binding in what is Scribble's main known function, the formation of cell junctions. This is largely a biochemical analysis with only a few in vitro assays added*.

In order to address the potential role of SCRIB-Spectrin function in cell junction formation we used a well-established “calcium switch” model that involves reversible disruption of epithelial junctions by extracellular calcium depletion followed by a rapid junctional reassembly triggered by calcium repletion (Gonzalez-Mariscal, Contreras, 1990). As previously described (Ivanov, Young, 2010) SCRIB-depleted cells show a significant delay in tight junction reassembly inCaco-2 cells. In these conditions, WT SCRIB partially rescues the phenotype while the SADH mutant does not (new Figure 4—figure supplement 3). This observation suggests that SADH domains are involved in cell junction formation.